



# Water partitioning in a Neotropical Savanna forest (Cerrado *s.s.*): interception responses at different time-scales using adapted versions of the Rutter and the Gash models

Lívia Rosalem[1], Miriam Coenders-Gerrits [2], Jamil. A. A. Anache[3], Seyed M. M. Sadeghi[4,5], Edson Wendland[1]

[1] University of São Paulo, Department of Hydraulics Engineering and Sanitation, São Carlos, 13566-590, Brazil
[2] Delft University of Technology, Water Resources Section, Delft, 2628 CN, the Netherlands
[3] Federal University of Mato Grosso do Sul, Campo Grande, 79070-900, Brazil
[4] Department of Forest Engineering, Forest Management Planning and Terrestrial Measurements, Faculty of Silviculture and Forest Engineering, Transilvania University of Brasov, Şirul Beethoven 1, 500123, Brasov, Romania
[5] School of Forest Resources and Conservation, Newins-Ziegler Hall, University of Florida, Gainesville, Florida, 32611, USA

*Correspondence to*: Miriam Gerrits-Coenders (a.m.j.coenders@tudelft.nl) and Lívia Rosalem (liviarosalem@gmail.com)

**Abstract.** Cerrado is the broadest Savanna ecosystem of South America and has an important role in our global climate. How rainfall finds it way through the vegetation layers of the undisturbed Cerrado forest is of utmost importance to understand the evaporation process and the water availability in this unique ecosytem. Nonetheless, only few studies consider the partitioning of rainfall in the Cerrado. And if they do, these studies are limited by only considering interception by the canopy, while the forest floor can intercept a significant amount as well. Additionally, the studies often apply canopy interception models that were calibrated on short term monitoring. Hence evaluating how interception models perform at different time-scales and how the interception process responds to seasonal changes is poorly understood for the Cerrado forest. In this study we aimed to evaluate the canopy and forest floor interception estimates at different time-scales and its seasonal response for an undisturbed Cerrado *s.s.* forest in Brazil. Two commonly used interception models (Rutter and Gash) were adapted to include forest floor interception using observations of both canopy and forest floor interception during a 32 months study period. Our results show that the models are suitable to estimate throughfall and infiltration at daily basis, but not the evaporative processes. We confirmed that both models had limitations to simulate very high interception rates on an event scale. Nonetheless, both models are able to reproduce the total interception well at monthly scale ($R^2$ = 0.7–0.97, NSE = 0.63–0.85), and they can represent seasonal trends in the interception process in Cerrado *s.s.* forests. Nevertheless, the Rutter model seems to perform better when seasonal parameters are used than the Gash model, but both models are equally valuable to inter-annual analysis when non-seasonal parameters are used.



## 1 Introduction

Tropical forests have an important role in mediating the global climate (Staal et al., 2020; Mitchard, 2018), and maintaining the atmospheric moisture in the air – indicated by the high interception vegetation ratios and its relation with high local evaporation recycling (Wang-Erlandsson et al., 2014). In these forests, 50–60% of annual rainfall is fed back to the atmosphere through the evaporation of intercepted water and transpiration (Loescher et al., 2005; Rocha et al., 2004). The great number of plant species and individuals per hectare in tropical forests reduce the homogeneity of the water cycle (Levia et al., 2020), which turns a challenge on quantifying the interception processes in such forests (Blyth and Harding, 2011).

The Brazilian Cerrado is a Neotropical tropical savanna covering 200 million hectares (one quarter of Brazil's terrestrial area) that contributes to 43% of Brazil's water surfaces outside the Amazon. Nowadays, despite its bio- and hydrological importance, as little as 19.8% of undisturbed vegetation cover remains (Strassburg et al., 2017). Being one of the largest farming frontiers of the world, understanding how water is partitioned in the remained undisturbed Cerrado areas are urged to evaluate the possible water balance trade-offs due to the potential land cover changes in such areas and the climate change impacts (Anache et al., 2019). Interception modelling is essential for evaluating the water balance, and plays an important function in hydrological simulations and quantifying ecohydrological services.

The interception process in forests occurs through the water retention on the forest canopy, trunk and forest floor, and its subsequential evaporation (Klamerus-Iwan et al., 2020). The residence time of retained water differs for each interception component, like few hours for canopy to several hours or days for forest floor (Gerrits et al., 2007; Grelle et al., 1997; Wang-Erlandsson et al., 2014). Studies about the interception process have been made in diverse forests around the world in order to quantify or better understand the involved processes (Aydın et al., 2018; Sadeghi et al., 2015; Grunicke et al., 2020; Jiménez-Rodríguez et al., 2020; Prasad Ghimire et al., 2017). Moreover, interception models have been used world widely not only as a tool to extrapolate the interception observations, but also to provide insights about process interactions, like rainfall intensity and canopy storage capacity, and the effects of biotic and abiotic factors onto the interception processes (Muzylo et al., 2009; Lopes et al., 2020). However, studies have pointed out that some current interception models presents limitations to simulate vary low and very high levels of interception on an individual event scale (Linhoss and Siegert, 2020).

The interception model by Rutter et al. (1971) and  by Gash (1979), and also their reformulated versions to sparse forests (Valente et al., 1997), are still the most applied interception models (Muzylo et al., 2009; Ma et al., 2019; Lopes et al., 2020). The Rutter model was the first physically-based and continuous model to represents the interception process (Rutter et al., 1971), which is essentially a dynamic water balance of rainfall input, storage and outputs as drainage and evaporation.

Gash (1979) presented an analytical interception model which is a simplification version of Rutter's model. Whilst considered some assumptions of linear regression equations, this model maintains the physical-based reasonings of the Rutter's model. For Gash model, rainfall input is assumed as discrete storm events on daily basis. In such storms, the interception process occurs through three distinct phases. Firstly, on the "wetting phase" the interception reservoirs start to be filled from the onset



rainfall to the start of the "saturated phase". This second phase is followed by the final "drying phase" in which the evaporation process is dominant (Valente et al., 1997; Muzylo et al., 2009).

Interception models were already applied to different forests/ecoregions in Brazil. For example Vieira and Palmier (2006) analyzed the Rutter and also Gash model to a remanent mixed forest of secondary semideciduous' forest and one kind of Cerrado forest, and also Sá et al. (2015b), and Junqueira et al. (2019a) to Atlantic forests, and Lloyd et al. (1988) and Cuartas et al. (2007) investigated to the Amazonic forest. However, all these studies focus only on the canopy interception process and disregard or indirectly estimate the forest floor interception (Coenders-Gerrits et al., 2020; Stan et al., 2017). Moreover, the

modeling results of Cerrado areas are often based on short term datasets, usually consisting of only one wet season (Távora and Koide, 2020). Hence evaluating how interception models perform at different time-scales and how the interception process responds to seasonal changes is poorly understood for the Cerrado forest. In this study we aimed to evaluate the canopy and forest floor interception estimates at different time-scales and its seasonal response for an undisturbed Cerrado *s.s.* forest in Brazil. Two commonly used interception models (Rutter and Gash) were adapted to include forest floor interception using

observations of both canopy and forest floor interception during a 32 months study period. The models' performance was evaluated on daily, biweekly and monthly basis and with seasonal and non-seasonal models' parameters. Thus, the models' responses at different time-scales and the responses to changes on forest stand and weather (i.e., through seasons) were evaluated.

## 2 Material and methods

### 2.1 Study area

The study site is a Cerrado *s.s.* forest in the Arruda Botelho Institute (IAB), located in Itirapina, State of São Paulo, Brazil (22°11'5'' S, 47°51'11'' W, Fig. 1), on an altitude of 763 m a.m.s.l., which ranges between 716 m and 802 m within the studied fragment (Farr et al., 2007) . This area (ca. 260 ha) is in the Basaltos do Paraná ecoregion (Sano et al., 2019) and is an important area for biological and hydrological surveys (outcrop zone of the Guarani Aquifer System) (Anache et al., 2019;

Reys et al., 2013). Consistent with local meteorological data between 1971 and 2014, the mean annual rainfall and temperature are 1486 mm yr$^{-1}$ and 21.6 °C, respectively (Cabrera et al., 2016). The area is in a subtropical region, a Cwa climate according to Köppen classification (Alvares et al., 2013), with a marked wet season (hot and humid) from October to March and dry season from April to September.

The Cerrado ecoregion presents different physiognomies, regarding the increasing density of woody species. Physiognomies

range from grasslands (Campo limpo) to dry forests (Cerradão) with intermediate types as savannas (Campo sujo and Campo cerrado) and woody savannas (Cerrado *stricto sensu*) (Alberton et al. 2019; Coutinho 1978).

The forest study area is characterized by a dominant woody layer reaching 6-8 m high with discontinuous crown cover and a kind of continuous herbaceous layer (Reys et al., 2013; Camargo et al., 2011). The vegetation is classified as semi-deciduous based on long-term leaf exchange strategies (Alberton et al., 2019; Camargo et al., 2018). The density of stand is 15,522 sph,



whereas the maximums DBH (diameter at breast height) and tree height values are 34.7 cm and 12 m, with abundant presence of *Bauhinia rufa* (Bong.) Steudel, *Xylopia aromatica* (Lam.) Mart., *Miconia rubiginosa* (Bonpl.) A. DC., *Virola sebifera* Aubl., and *Myrcia guianensis* (Aubl.) DC. (Reys et al., 2013) . In the study area, the average canopy openness of 29.7 ± 8.8 % and average PAR (photosynthetic active radiation) of 1041.8 ± 427.4 µmol.m$^{-2}$.s$^{-1}$ (Reys et al., 2013).

### 2.2. Experimental setting

This study comprises the monitoring period between June 01$^{st}$ of 2017 and February 06$^{th}$ of 2020. Precipitation, temperature, relative air humidity, wind velocity, net solar radiation, and others environmental variables were collected in the site each 10 minutes through a meteorological tower of 11 m of height. In case of missing data of the main monitoring weather station, we used the nearest available meteorological data from site 1 (Anache et al., 2019), which was 1.7 km away from our site.

**Figure 1**. Location of the study area (from ©Google Maps, 2022).

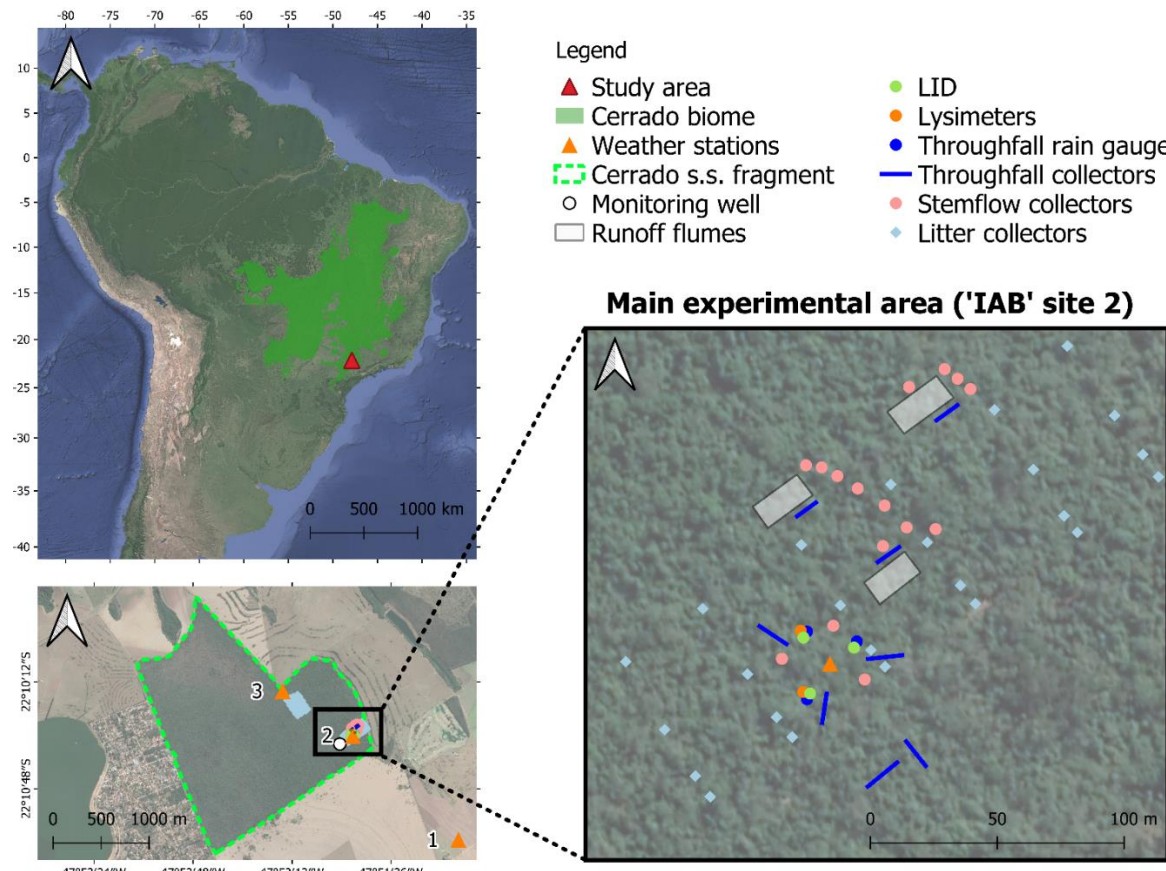


To investigate the interception process in the Cerrado *s.s.*, canopy, trunk and forest floor interception were measured. Canopy and trunk interception were indirectly determined by the difference between the rainfall ($P_g$) and throughfall ($T_f$ ) and stemflow





$(T_s)$, respectively. By including forest floor interception, the total forest interception is measured by the difference between the

$P_g$ and the infiltration $(F)$, as in Eq. (1).

$$E_{i,c} + \frac{dS_c}{dt} + E_{i,t} + \frac{dS_t}{dt} + E_{i,f} + \frac{dS_f}{dt} = P_g - F \qquad (1)$$

where $S_c$, $S_t$ and $S_f$ are the storage capacities (mm) of the canopy, the trunks, and the forest floor, respectively, and $E_{i,c}$ , $E_{i,t}$ , and $E_{i,f}$ are the evaporation from these components (mm) in a certain  period of time $(t)$.

$T_f$ was measured, also each 10 min, through four pluviographs of 0.254 mm resolution. Additionally, three gutters linked to

the pluviographs (tipping bucket resolution of 0.048 mm or 0.029 L) were used and five additional gutters that were directly connected to reservoirs for manual measurements after accumulated events (> 15 mm). For $T_s$ monitoring, we installed three automatic collectors. Plastic hoses were wrapped around six trees (two per each collector) at breast height that channeled the $T_s$ through pluviographs (tipping bucket volume of 5 mL) to reservoirs. In addition, $T_s$ was measured through more 12 manual collectors. All trees were selected according to the DBH. More details about the equipment are given in the Table 1.

Forest floor interception and infiltration were measured each 10 min through LIDs – Litter Interception Devices (Rosalem et al., 2019). We installed three of these weighing LID devices, but due to problems with a load cell only the interception records of two LIDs were used. The LIDs were installed close to the meteorological tower and were filled with quasi-undisturbed litter samples of the thickness of about 6 cm. The forest litter samples were changed each August considering its high decomposition rate in dense Cerrado areas (half-life for the decomposing material around 1.8 year) (Cianciaruso et al., 2006).


**Table 1**. Details about monitoring instrumentation in the Cerrado's study area (site 2) and in another meteorological station (site 1).

| Variable (unit) | Sensor (model) | Total quantity | Height/ or depth*(m) | Monitoring station site |
|---|---|---|---|---|
| Temperature (°C) and relative humidity (%) | Thermo hygrometer (HMP45C) | 2 | 2 and 11 | 1 and 2 |
| Precipitation (mm 10 min$^{-1}$) | Tipping bucket rain gauge (Hydrological Services TB4) | 2 | 2 and 11 | 1 and 2 |
| Net radiation (W m$^{-2}$) | Net radiometer (Kipp & Zonen NR-LITE2) | 1 | 11 | 2 |
| Solar radiation (W m$^{-2}$) | Radiometer (Kipp & Zonen CMP3) | 1 | 2 | 1 |
| Wind direction and velocity (m s$^{-1}$) | Anemometer (Young 05103) | 2 | 2 and 11 | 1 and 2 |
| Throughfall (mm 10 min$^{-1}$) | Gutter – manual measures (6 m x 0.1 x 0.1 m) | 5 | 0.6 | 2 |
| | Gutter – automatic measures (6 m x 0.1 x 0.1 m) | 3 | 0.6 | 2 |
| | Tipping bucket rain gauge (Davis Instruments 7857) | 4 | 1.5 | 2 |
| Stemflow (mm 10 min$^{-1}$) | Manual collectors | 12 | - | 2 |
| | Automatic collectors | 3 | - | 2 |





| | | | | |
|---|---|---|---|---|
| Forest floor interception (mm 10 min[-1]) | LID | 2 | 0 | 2 |
| Infiltration (mm 10 min[-1]) | LID | 3 | 0.1 * | 2 |

## 2.3 Interception models

We applied the Rutter model (Rutter et al., 1971, 1975; Rutter and Morton, 1977) that was adapted by Gerrits et al. (2010), and developed an adapted version to the original Gash model (Gash, 1979) to include forest floor interception. For both models, we intended to maintain the original model concepts as well as its original assumptions. However, we extended the models by differentiating between canopy and forest floor interceptions.

The storage capacities parameters were determined following the procedures by Rutter et al. (1971) and Gash and Morton
(1978). They are determined by the negative intercept of the upper envelop line of a regression between observed values of an input process (e.g., $P_g$) and its subsequent process (e.g., $T_f$) (Robins, 1969, apud (Rutter et al., 1971)). Thus, the $S_c$ and $S_t$ parameters were determined by the linear regression between $P_g$ and $T_f$ records, and between $P_g$ and $T_s$ records, respectively (Rutter et al., 1971; Gash and Morton, 1978). By the same procedure, the storage capacity of the forest floor ($S_f$) can be determined through the linear regression lines between $T_f$ and $F$ values. These regression analyses were carried out based on
108 independent rain storms (each one preceded by at least 24 hours without rainfall) out the total of 236 rain days during the calibration period.

### 2.3.1 The Rutter model adapted

Here, the adapted Rutter model applied by Gerrits et al. (2010), which include the forest floor interception reservoir as well, was used with some adjustments. An overview of the model structure can be found in Fig. 2.
The model was run at 10 minute intervals and the potential evaporation ($Ep$) was obtained through the Penman equation (Penman and Keen, 1948; Rutter et al., 1971). As some visual field observations in situ indicated it would be necessary many days for the complete evaporation from the trunks, we assumed here $\epsilon$ equal to the lowest value, 0.01 (Rutter and Morton, 1977; Valente et al., 1997). Moreover, as some studies have indicated that evaporation during the rainfall is underestimated by the Penman-Monteith equation with canopy resistance set to zero (Saito et al., 2013; van Dijk et al., 2015), the evaporation
process was prioritized over the others process on each running step.

Observed $T_f$ records showed recurrent high intensity rainfall inputs at 10 min time scale (102 records with rainfall ≥ 6 mm, i.e., rainfall intensity ≥ 36 mm h[-1]). These observations have indicated that apparently some dynamic storage on the canopy ($C_c$) is present. Also, Klaassen et al. (1998) observed by using microwave transmission to measure the water storage on the canopy, that during rain events the dynamic storage is affected by rain intensity. The results of Aston (1979) and Lloyd et al.
(1988) also agree with this, and the authors recommended, as Valente et al. (1997), that it should prevent a build-up of water





on the canopy. Therefore, we added a maximum dynamic storage parameter ($C_{cmax}$) as a threshold to the amount water retained on the canopy before the end of the rainfall storm.

The drainage process ($D$) was modelled as a threshold process during excessive events ($C_c \geq C_{cmax}$), as $D = C_c - C_{cmax}$. If the dynamic storage is less than $C_{max}$, the drainage was modelled as an exponential equation (Rutter et al., 1971). Even when 160 $C_c < S_c$, drainage could still happen through the shake off of rain drops (Gerrits et al., 2010) and was modelled by the same exponential equation. The threshold values to the exponential equation coefficients $D_s$ and $b$ were determined by running a water balance on the canopy under the dynamic storage condition of $C_c < S_c$. After, these coefficients were calibrated by assuming those maximum values as their threshold.

**Figure 2.** Overview of the Rutter model with the forest floor interception included.

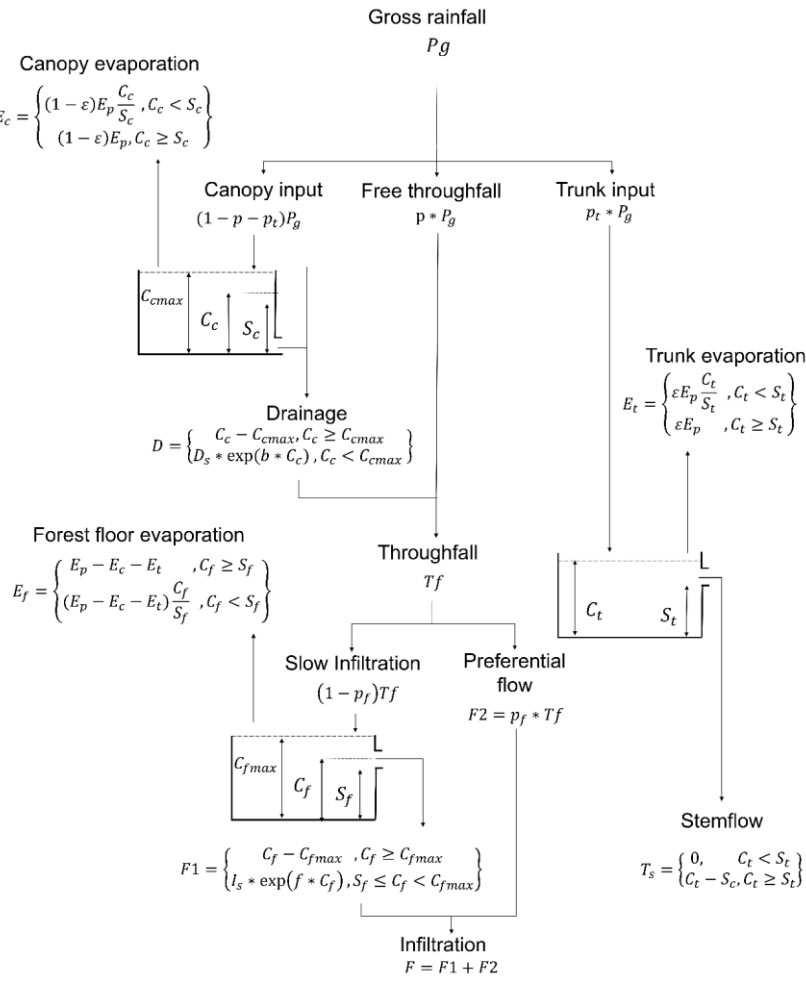

The drainage, together to the free throughfall ($p$), turn into $T_f$ and reaches the forest floor. The forest floor reservoir is modelled similarly to the canopy. Considering that $T_s$ contributes few to the forest floor interception, and also that our forest floor





interception measures have not included $T_s$ inputs, $T_f$ was considered the only input to forest floor. The free infiltration
parameter, $p_f$, refers to part of the $T_f$ that passes through the forest floor by the preferential ways, while part of the remaining
water $(1 - p_f)$ takes more time to passes through. At the same time the forest floor interception increases, the infiltration rate
increases until a certain threshold $(C_{fmax})$. $C_{fmax}$ is the maximum storage capacity of the forest floor (Rosalem et al., 2019;
Sato et al., 2004) and above this threshold, all input would be rapidly converted to infiltration $(C_f - C_{fmax})$.

$S_f$ represents the water that will not turn into infiltration, i.e., the storage without the gravitational water (Sato et al., 2004;
Dunkerley, 2015). As it is conceptual similar to $S_c$ and $S_t$, it was obtained following the same procedure to obtain canopy and
trunk storage capacities. Otherwise, $C_{fmax}$ was obtained based on the observed forest floor interception data. As different
forest litter samples were used along the entire monitoring period, the $C_{fmax}$ was determined by the average value among the
higher water content peaks observed. A similar exponential equation used to the drainage process was applied to the infiltration
process while $C_f \leq C_{fmax}$ and $C_f > S_f$. The exponential equation coefficients $Is$ and $f$ were obtained as the $D_s$ and $b$ of the
drainage process.

### 2.3.2 The Gash model adapted

Admitting the same assumptions of the original Gash model (Gash, 1979), mean throughfall rate $\overline{T_f}$ is calculated like $\overline{R}$
(Valente et al., 1997; Gash, 1979) using the observed throughfall measurements. As for canopy reservoir, the water amount to
saturate the forest floor $(T'_f)$ was determined using the forest floor structure parameters $p_f$, and $S_f$ previous determined, as
shown in Eq. (2).

$$T'_f = -\frac{\overline{T_f}}{\overline{E}} S_f \, ln\left[1 - \frac{\overline{E}}{(1 - p_f)\overline{T_f}}\right] \qquad (2)$$

Also, similarly to canopy modeling the forest floor interception was modeled regarding to throughfall inputs per rain day. The
$v$ rain days refer to days in which the saturation condition is achieved $(T_f \geq T'_f)$ whereas the $z$ rain days refer to rainfall events
where saturation is not reached $(T_f < T'_f)$. A summary of the equations used for $v$ and $z$ rain days is given in Table 2.

**Table 2**. Equations to forest floor interception estimation for each rainy day's condition.

| Rain day condition | Forest floor interception |
|---|---|
| $z$ rainy days (small storms insufficient to saturate the forest floor) | $(1 - p_f) \sum_{j=1}^{z} T_{fj}$ |
| $v$ rainy days (storms large enough to saturate the forest floor) | $(1 - p_f) \sum_{j=1}^{v} T'_{fj} + \frac{\overline{E}}{\overline{T_f}} \sum_{j=1}^{v} (T_{fj} - T'_{fj})$ |



The infiltration ($F$) (mm) is calculated (Eq. (3)) alike $T_f$ (mm), and the total interception is estimated by adding the forest floor interception by Eq. (4):

$$F = p_f \sum_{j=1}^{v+z} T'_{fj} + \left[ (1 - p_f) - \frac{\bar{E}}{\bar{\bar{T}}_f} \right] \sum_{j=1}^{v} (T_{fj} - T'_{fj}) \qquad (3)$$

$$I = (1 - p - p_t) \sum_{j=1}^{n} P'_{g,j} + \frac{\bar{E}}{\bar{R}} \sum_{j=1}^{n} (P_{g,j} - P'_g) + (1 - p - p_t) \sum_{j=1}^{m} P_{g,j} + q S_t + p_t \sum_{j=1}^{m+n-q} P_{g,j}$$

$$+ (1 - p_f) \sum_{j=1}^{z} T_{fj} + (1 - p_f) \sum_{j=1}^{v} T'_{fj} + \frac{\bar{E}}{\bar{\bar{T}}_f} \sum_{j=1}^{v} (T_{fj} - T'_{fj}) \qquad (4)$$

where $P'_g$ (mm) is the amount of water necessary to saturate the canopy and $m$, $n$ and $q$ refer to the rainy days that saturation phase is reached or not for canopy or trunks from the original version of Gash model (Gash, 1979). Equations of the original Gash model can be see in the Appendix A (Tab. A1) and also in Valente et al. (1997).

### 2.4 Statistical analysis

We split the dataset into two parts. From June of 2019 to May of 2019, we used to calibrate our Rutter and Gash models, while
the second part, from June of 2019 up to of 07th February of 2020, was used for validation period. The observed and modeled data were analyzed through Spearman correlations. Quantitative approaches were used to analyze the model performances as root mean squared error (RMSE), mean bias error (MBE), normalized mean error (NME), and the Nash-Sutcliffe coefficient (NSE) (Gupta et al., 2009). The performances were evaluated on daily, biweekly and monthly time-scales, by aggregation of the results. Because the Gash model is applied at daily time-scale, the Rutter model results were evaluated at a minimum on a
daily time-scale, thereto the model's results could be compared.

### 3. Results and discussion

During the 981 days monitored, 319 days were rainy days and in total 3824 mm of rain was recorded. Most of the rain events (62 %) had a small storm magnitude (< 10 mm), but it accounted for only 15 % of the total rainfall input. Whilst rain days with more than 30 mm of rain (11 % of the rain days) represented 43 % of the volume. Cumulative $P_g$ ranged from 0.25 mm day$^{-1}$
up to 82.05 mm day$^{-1}$, and the mean daily $P_g$ was 10.1 ($\pm$13.1 mm day$^{-1}$). The maximum rainfall intensity observed at 10 minutes was 17.5 mm 10 min$^{-1}$ (mean = 0.8 $\pm$1.4 mm 10 min$^{-1}$). Although being shorter, the validation period presented proportionally more intense storm events than the calibration period, which accounted for 35 % of the total rainfall monitored in this study (see in Fig. 1 in the Appendix B).

$Ep$ was calculated each 10 minutes through Penman-Monteith equation with crop resistance set to be zero (Gash, 1979; Rutter
et al., 1975; Valente et al., 1997). The maximum potential evaporation is 1.1 mm h$^{-1}$ and the mean is 0.2 mm h$^{-1}$. The mean





evaporation rate during storms ($\bar{E}$) ranged from 0.02 to 0.09 mm h$^{-1}$ which was similarly low as what was estimated by Cabral et al. (2015) (mean = 0.08 mm h$^{-1}$) to a Cerrado *stricto sensu* forest also in the Southeast of Brazil. The maximum temperature observed was 36.7° C and the mean was 21.3° C. The observed relative air humidity and net radiation corresponded on average to 66.8 (± 20 %) and 154.6 (± 239 W m$^{-2}$), respectively (Fig. 3).

**Figure 3**. Daily potential evaporation (mm day$^{-1}$) and solar radiation (W m$^{-2}$) observed on the study site.

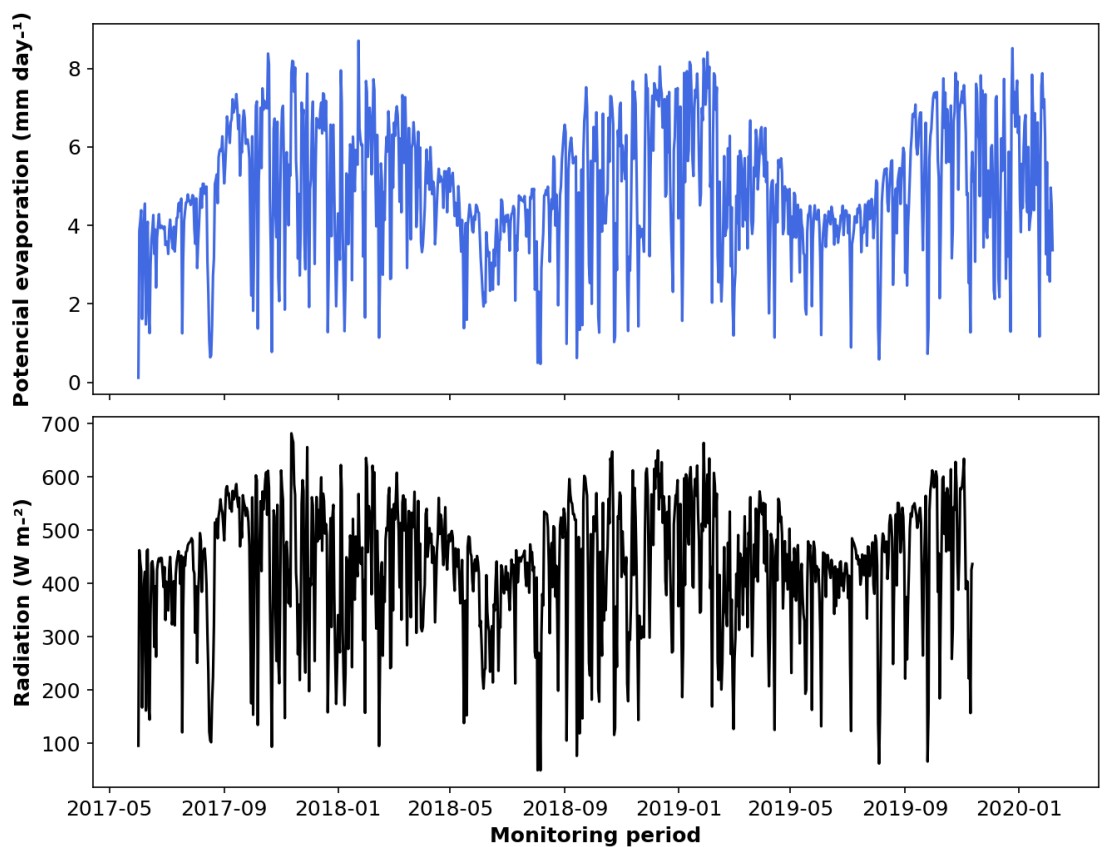

Sá et al. (2015a) verified the Rutter and Gash models' performances to a mixed ombrophilous forest in Brazil and they found better total interception results when the model parameters were calibrated. The calibrated $S_c$ value obtained by them (3.96 mm) is more similar to the empirical $S_c$ values found in this study for Cerrado *s.s.* (Table 3).

**Table 3**. Non-seasonal ecohydrological parameters to the Rutter and the Gash models.

| Parameter | Value | Parameter | Value |
|---|---|---|---|
| $S_c$ (mm) | 3.8 * | $C_{cmax}$ (mm) | 7.6 |
| $p$ (unitless) | 0.35 * | $C_{fmax}$ (mm) | 5.25 |
| $S_f$ (mm) | 1.8 ** | $Ds$ (mm 10 min$^{-1}$) | 0.001 |
| $pf$ (unitless) | 0.45 ** | $b$ (mm$^{-1}$) | 0.38 |
| $S_t$ (mm) | 0.015 *** | $Is$ (mm 10 min$^{-1}$) | 0.001 |
| $pt$ (unitless) | 0.004 *** | $f$ (mm$^{-1}$) | 0.38 |
| $\epsilon$ (unitless) | 0.01 | | |





## 3.1 Throughfall

From the total 730 days of the calibration period, 14 % (104 days) out are gaps in $T_f$ records due to problems with the sensors. We found $T_f$ was on average 72 % of $P_g$. During calibration and validation periods, the $T_f$ represented 70 % and 72 % of $P_g$, respectively.

These values are greater those founded by Vieira and Palmier (2006) (67 %) to a mixed forest of Cerrado and Ombrophilous forest in Brazil. The values are lower than the values founded by Honda and Durigan (2016) to a Cerrado *s.s.* forest (75.4 %), and by Távora and Koide (2020) (75.3 %) to a riparian forest in Cerrado. However, the found percentage is close to the found by Lima and Nicolielo (1983) to a Cerradão forest (72.7 %), and by Oliveira (2014) (72.7 %) to a riparian forest in Cerrado. The throughfall values were also lower than those found out to Amazon forest (Cuartas et al., 2007) and to Atlantic forest (Sá

et al., 2015b; Sari et al., 2015, 2016). Nevertheless, the observed values are corroborated by the infiltration monitored (forward presented here).

Both models were capable of modelling the throughfall well on a daily time scale for both the calibration ($R^2$ = 0.91–0.96) as the validation period ($R^2$= 0.94–0.98). The Gash model underestimated the total $T_f$ in 5.4 mm (-0.7 % of observed) during calibration and overestimated $T_f$ in 103.5 mm (+17.3 % of observed) during the validation phase. Similarly, the Rutter model

overestimated during the validation, 68.7 mm (+11.5 % of observed), but also overestimated during the validation period, in 10.4 mm (+1.3 % of observed) (Table 4). Scatter and accumulative plots of modelling results are presented in the Appendix C. The quantitative metrics indicated that the Gash model had better performance than Rutter's model on a daily time-scale, for both calibration (NSE = 0.92) and validation (NSE = 0.94) periods, with lower RMSE and NMPE values and slightly greater MBE value during validation period (Table 5). However, for biweekly and monthly time-scale analysis, both models

had near NSE, RMSE and MBE with lower NMPE values by the Rutter model than by Gash model.

**Table 4.** Accumulated values of the Rutter and Gash models with non-seasonal parameters during calibration and validation periods.

| Process | Period | Gash model adapted | | Rutter model adapted | |
|---|---|---|---|---|---|
| | | Modeled (mm) | Modeled error (%) | Modeled (mm) | Modeled error (%) |
| $T_f$ | Calibration | -5.42 | -0.7 | +10.40 | +1.3 |
| | Validation | +103.50 | +17.3 | +68.74 | +11.5 |
| $F$ | Calibration | -25.35 | -2.4 | -56.27 | -5.4 |
| | Validation | +182.29 | +24.1 | +109.61 | +14.5 |
| $T_s$ | Calibration | -1.69 | -48.9 | -1.60 | -46.5 |
| | Validation | -1.77 | -44.6 | -1.91 | -47.9 |
| $E_f$ | Calibration | -42.87 | -14.2 | +2.74 | +0.9 |
| | Validation | -13.47 | -11.7 | +9.78 | +8.5 |
| $E_c$ | Calibration | -83.63 | -22.8 | -158.70 | -43.4 |
| | Validation | -115.94 | -42.5 | -153.01 | -56.1 |
| $I$ | Calibration | -34.05 | -5.2 | +29.90 | +4.5 |
| | Validation | -210.52 | -39.3 | -166.67 | -31.1 |





**Table 5.** Quantitative metrics and performance analysis of the Rutter and Gash models with non-seasonal parameters on daily (D), biweekly (B) and monthly (M) time scales to calibration and validation periods. The best values for each metric are highlighted in bold (calibration) and italic (validation).

| Process | Time scale | Period | Gash model adapted | | | | | Rutter model adapted | | | | |
|---|---|---|---|---|---|---|---|---|---|---|---|---|
| | | | RMSE | MBE | NMPE | NSE | R² | RMSE | MBE | NMPE | NSE | R² |
| | | | (mm [time scale]⁻¹) | | (%) | (unitless) | (unitless) | (mm [time scale]⁻¹) | | (%) | (unitless) | (unitless) |
| $T_f$ | D | Calibration | **1.26** | **+0.06** | 29 | **0.92** | 0.96 | **1.74** | **+0.12** | 37 | 0.85 | 0.91 |
| | | Validation | *2.05* | *+0.43* | 25 | *0.94* | 0.98 | *2.25* | *+0.36* | 29 | 0.92 | 0.94 |
| | B | Calibration | 14.24 | -8.74 | 11 | 0.91 | **0.98** | 10.67 | -4.78 | **10** | **0.91** | 0.98 |
| | | Validation | 9.93 | +4.86 | 17 | 0.94 | *0.99* | 6.78 | +2.97 | *11* | *0.97* | 0.98 |
| | M | Calibration | 14.87 | +5.82 | **15** | 0.88 | 0.98 | 14.83 | +7.06 | 13 | 0.89 | **0.98** |
| | | Validation | 20.85 | +15.70 | *14* | 0.92 | 0.99 | 12.34 | +7.97 | 8 | 0.97 | *0.99* |
| $F$ | D | Calibration | **1.87** | **-0.03** | 37 | 0.84 | 0.91 | **2.06** | **-0.08** | 45 | 0.81 | 0.85 |
| | | Validation | *3.30* | *+0.73* | 38 | 0.82 | 0.97 | *2.95* | *+0.43* | 36 | 0.86 | 0.93 |
| | B | Calibration | 10.12 | +0.58 | 22 | 0.84 | 0.88 | 8.61 | +0.03 | 22 | 0.85 | 0.88 |
| | | Validation | 17.71 | +9.23 | 26 | *0.89* | 0.99 | 13.26 | +5.73 | 19 | 0.94 | 0.98 |
| | M | Calibration | 11.99 | -2.93 | **15** | **0.91** | **0.92** | 12.00 | -4.16 | **15** | **0.91** | **0.92** |
| | | Validation | 27.05 | +19.72 | *21* | 0.87 | *0.99* | 18.55 | +11.96 | *13* | *0.94* | *0.99* |
| $T_s$ | D | Calibration | **0.02** | **-0.01** | 57 | **0.63** | 0.79 | **0.02** | **-0.01** | 52 | **0.64** | **0.82** |
| | | Validation | *0.04* | *-0.01* | 54 | 0.56 | 0.67 | *0.04* | *-0.01* | 57 | 0.42 | 0.48 |
| | B | Calibration | 0.12 | -0.07 | 56 | 0.46 | 0.53 | 0.08 | -0.03 | 54 | 0.51 | 0.59 |
| | | Validation | 0.21 | -0.09 | 48 | 0.53 | *0.82* | 0.24 | -0.10 | 52 | 0.41 | 0.70 |
| | M | Calibration | 0.22 | -0.18 | **48** | 0.42 | 0.69 | 0.13 | -0.06 | **45** | 0.48 | 0.73 |
| | | Validation | 0.26 | -0.15 | *22* | *0.73* | 0.82 | 0.17 | -0.15 | *23* | *0.70* | *0.78* |
| $E_f$ | D | Calibration | **0.83** | **-0.06** | 140 | -5.75 | 0.03 | **0.72** | **+0.00** | 108 | -4.12 | 0.19 |
| | | Validation | *0.96* | *-0.05* | 154 | -5.55 | 0.01 | *0.83* | *+0.04* | 118 | -3.80 | 0.17 |
| | B | Calibration | 3.34 | -0.76 | 44 | 0.09 | 0.55 | 3.05 | +0.02 | 42 | 0.25 | 0.66 |
| | | Validation | 4.58 | -0.75 | 59 | -0.63 | 0.29 | 3.46 | +0.47 | 43 | 0.07 | 0.60 |
| | M | Calibration | 5.41 | -1.78 | **34** | **0.42** | **0.67** | 5.20 | +0.11 | **28** | **0.46** | **0.73** |
| | | Validation | 5.79 | -1.58 | *41* | *0.43* | 0.58 | 4.25 | +1.00 | *23* | *0.69* | *0.87* |
| $E_c$ | D | Calibration | 6.62 | -1.56 | 72 | 0.20 | 0.44 | 6.88 | -1.47 | 100 | 0.04 | 0.01 |
| | | Validation | *9.08* | *-2.31* | 73 | 0.14 | 0.54 | *9.59* | *-2.23* | 97 | 0.01 | 0.02 |
| | B | Calibration | **6.35** | -0.66 | 27 | 0.72 | 0.64 | **4.76** | **-0.17** | 25 | **0.72** | **0.73** |
| | | Validation | 10.68 | -4.99 | 39 | 0.59 | *0.87* | 10.31 | -4.25 | 37 | 0.62 | 0.82 |
| | M | Calibration | 11.58 | **+0.30** | **25** | **0.74** | **0.74** | 13.50 | +1.42 | 33 | 0.65 | 0.68 |
| | | Validation | 17.53 | -9.65 | 26 | *0.70* | 0.77 | 13.89 | -6.10 | *24* | *0.82* | *0.81* |
| $I$ | D | Calibration | **5.05** | -0.72 | 60 | 0.36 | 0.48 | **6.23** | -0.62 | 118 | 0.02 | 0.02 |
| | | Validation | *6.02* | *-1.22* | 56 | 0.33 | 0.57 | *7.30* | *-1.03* | 113 | 0.01 | 0.01 |
| | B | Calibration | 8.02 | -1.08 | 26 | 0.76 | 0.76 | 6.33 | **-0.14** | 27 | **0.80** | **0.80** |
| | | Validation | 19.42 | -10.80 | 42 | 0.60 | 0.92 | 20.26 | -8.97 | 40 | 0.56 | 0.79 |
| | M | Calibration | 10.08 | **+0.38** | **19** | **0.85** | **0.85** | 12.73 | +1.99 | **25** | 0.77 | 0.78 |
| | | Validation | 28.10 | -20.05 | *30* | *0.63* | *0.97* | 24.15 | -13.12 | *23* | *0.73* | *0.89* |



### 3.2 Canopy interception

The observed canopy interception indicated that the process corresponds on average to 33 % of the rainfall. On 38 rain events out of 172 events in which canopy evaporation ($E_c$) was observed, the total daily $E_c$ was greater than 6 mm day$^{-1}$. The

maximum ($P_g - T_f - T_s$) observed was 23.1 mm day$^{-1}$ and 10 rain events had $E_c$ greater than 12 mm day$^{-1}$. Although this is a very high number, studies have pointed out uncertainties over the controls on the downward heat flux, local horizontal advection, the possible upwards transport of small splash rainfall droplets (van Dijk et al., 2015) that can have important effect on the canopy interception process.

The observed canopy evaporation was 33 % of the total rainfall during the calibration period and 27 % during the validation

period. These values were calculated as precipitation minus the observed throughfall and stemflow by day. To verify these values, we calculated $E_c$ as a residual of the water balance for the entire period using other observed data, such as rainfall, infiltration, stemflow, and the forest floor evaporation ($E_c = P_g - F - T_s - E_f$) which resulted an average of 28 % of total rainfall. These values indicate that canopy evaporation based on the direct throughfall measures, with the fixed throughfall gauges (gutters and rainfall tipping buckets), could be overestimated up to 6 %.

As the forest floor evaporation should not be analyzed at a daily time-scale (see item "Forest floor interception"), the models' performance to estimate the canopy evaporation on different time scales were analyzed using the $E_c$ indirectly observed by using the automatic throughfall records.

Both models always underestimated the accumulate canopy evaporation, but the Gash model underestimated less than the Rutter model. During calibration, Rutter model underestimated the total $E_c$ values in -43 % and the Gash model in -23 %.

During the validation period, the $E_c$ values were more underestimated than during calibration by Rutter and Gash models, -56 % and -42 %, respectively. At daily time-scale, both models had bad performances (NSE < 0.20 and R² ≤ 0.54), but with a considerable improved performance for both models if the analysis is at biweekly or monthly time scales, e.g., NSE (0.59–0.82) and R² (0.64–0.87).

Through analysis of five main interception models, Linhoss and Siegert (2016) pointed out that canopy storage capacity is one

of the most important models' parameters, together with precipitation, solar radiation and rainfall duration. Their results also indicated that the canopy storage and the subsequent evaporation can be greater than those interception models could simulate, as observed here. Also, studies that used different approaches have found greater canopy storages, like Friesen et al. (2008) (> 6 mm) and Iida et al. (2017) (up to 7.2 mm), which highlight the challenges regarding empirical determination of canopy storage capacity.

Aside the limitations of both models to modeling picks of interception, by adding Ccmax in the Rutter model allowed to delay the drainage process and to retain more water on the canopy during the rainfall. However, due to low $Ep$ values during the rainfall and to the constraint evaporation factor ($Ep \cdot \frac{C_c}{S_c}$) it took in general more than 2 days to completely dry the canopy. For



this reason, the Rutter model did not perform well on daily time-scale as a high $S_c$ value was used. By using the average rate factor ($\bar{E}/\bar{R}$), the Gash model reduced the mentioned problems of canopy evaporation on Rutter's model and had better

performance on daily time-scale. But the main Gash model assumptions that there is only one storm per rainy day and that there is enough time for interception components to dry between each rainy day, can lead to overestimations and unrealistic values on daily time-scale.

### 3.3 Stemflow and trunk interception

For the water balance of forests, $T_s$ is usually the lowest volume component (see reviews of $T_s$ value by Van Stan and Gordon

(2018) and Sadeghi et al. (2020)). Our monitoring results indicated the same magnitude (0.3 % of $P_g$) as also founded similarly by Honda and Durigan (2016) and by Oliveira et al. (2014). The models had very similar performance for both calibration and validation periods. For the calibration, both models underestimated the total stemflow by 1.7 mm (-48.9 %) and 1.6 mm (-46.5 %), by Gash and Rutter models, respectively. Similarly, for validation period, the two models underestimated the total volume with a slightly increased, -48 % by Rutter model and decreased, -45 % by Gash model.

Despite the relative high underestimations, the predicted difference is too small, i.e., less than 2 mm. It should be mentioned that the total stemflow volume on validation was greater (0.38 % of $P_g$) than to calibration (0.33 % of $P_g$). Regarding to stemflow modeled, both models had good performances in validation at monthly time-scale, with NSE greater than 0.70. At daily time-scale, the NSE value were between 0.42–0.64, with high NMPE values ($\geq$ 52 %). Comparing them, Rutter's model had a slightly better performance for the calibration period while Gash's model had to validation period.

Gonzalez-Ollauri et al. (2020) found that stemflow yield is related to the tree canopy structure and also that thin trunks and small crowns can increase not just the stemflow yield but also the funneling process. In Cerrado forests the stemflow is more efficient to capture and channeling rainwater down to tree's base in smaller trees with compact crowns (Honda et al., 2015). However, Cerrados' trees do not seem to be adapted to drive a huge amount of rainwater to their bases (Honda et al., 2015). Instead, Cerrados' forests used other adaptation strategies, like a deep root system (Anache et al., 2019; Leite et al., 2018) and

a strong surface control correlated to leafless periods to optimize their water use (Oishi et al., 2010; Cabral et al., 2015). Particularly to Cerrado's forests, the basal area seems to be the most related parameter to stemflow (Honda and Durigan, 2016) and the groundwater depth and soil properties have a strong effect to woody structure in such forests (Leite et al., 2018; Honda and Durigan, 2016; Durigan and Ratter, 2006). Recently, Tonello et al. (2021) analyzed the stemflow from 36 Cerrado's trees species and they found that stemflow in Cerrado's trees is highly variable, and it affected by bark texture, DBH and canopy

geometry. Contrary to some previous studies in other forests, they found that tree's attributes may be more relevant than the density of individuals to total stemflow yield in Cerrado. Considering the effect of rainfall intensity on stemflow production (Dunkerley, 2014), field studies that analyze this effect regarding to the tree's characteristics, as woody structure and basal area, could shed more light to how this process occurs along the successional physiognomic gradients of Cerrado.

Total trunk evaporation accounted for less than 5 mm for calibration and validation periods. As there are no observed field

values to compare or to evaluate the models' performances, and the values are quite insignificant related to rainfall input, we





have not discussed these results here. Although, the trunk evaporation was taking into account for the total interception analysis.

## 3.4 Infiltration

During the calibration and validation periods infiltration amount represented 61.9 % and 58.5 % of the total rainfall,
respectively. Based on daily total throughfall and infiltration data, the Spearman correlation coefficient (0.94) indicated a strong correlation between the monitored values, and it is corroborated by the Kendall's tau correlation value of 0.83 to Tau-b statistic, with significant probability of dependence through significance test with α value of 0.05 (p-value ≤ 5,59.$10^{-33}$).

Like throughfall, infiltration was overestimated by both models during validation, but it was underestimated for both models during the calibration. As the results are not independent, throughfall modeled results may affect the infiltration and hence
forest floor interception. During the calibration period, the Rutter model outcome was 56.3 mm (-5.4 %) less than observed while the Gash model underestimated 25.3 mm (-2.4 %). The outcomes during validation were more different from the observations, with the more overestimation by the Gash model (+24.1 %, equal to 182.3 mm) than by the Rutter model (+14.5 %, equal to 109.6 mm).

Both models well performed at daily time-scale, with a slightly better performance of the Gash model, which presented lower
RMSE, MBE and NMPE values. However, at biweekly and monthly time-scales the Rutter model performed better than the Gash model (greater NSE values), mainly during validation period.

These overestimates during validation can be partially explained by the modeled throughfall overestimation during the validation. Aside these overestimated values, both models had a good performance, even at daily time-scale (NSE ≥ 0.81).

## 3.5 Forest floor interception

On average, the observed forest floor evaporation was  12 % of annual rainfall. The Rutter model had better values than the Gash model during calibration, 304.1 mm (+0.9 %) and 258.5 mm (-14.2 %), and on validation, 124.3 mm (+8.5 %) and 101.1 mm (-11.7 %), respectively. The overestimated throughfall during validation may have an effect on the modeled forest floor interception. It could partially explain the increased overestimation by the Rutter model.

Both models had bad performances on a daily time-scale, but on a monthly scale both models had considerable increased
performances, being the Rutter model the best one, i.e., lower MAE, RMSE, MBE and NMPE. At biweekly time-scale the modeled values still had not a good relation to the observed values, but at least the Rutter model presented more clear tendency (R² ≥ 0.60).

Due to the average rate factors ($\bar{E}/\bar{R}$), the Gash model could simulate higher values than the Rutter model during the rain. In our adapted Rutter model, the canopy and trunk evaporation are prioritized over the forest floor evaporation. As the $Ep$ below
the canopy is lower than above, and due to different weather conditions near the understory (Coenders-Gerrits et al., 2020). In addition, the forest floor and the canopy differ in physical structure, resource availability and biotic conditions (Yanoviak and



Kaspari, 2000), which imply different energy and water fluxes. Hence, the modelling of canopy evaporation during the rainfall was dominant over forest floor evaporation, which caused delays onset on the last one.

Although the adapted Gash model could simulate greater forest floor evaporation during the rain, the main problem relied on
the assumption of there is enough time between the rains to dry the forest floor as well as for the canopy. Despite this assumption is not a big problem to the canopy evaporation, for which the evaporation should take only hours (Gerrits et al., 2010; Klamerus-Iwan et al., 2020; Rutter and Morton, 1977), is to the forest floor since the observed evaporation showed that the process takes more than days – which was more plausible during the dry season.

The estimated $S_f$ value for the study area, 1.8 mm, is in range with values from literature (0.6–8.0 mm) regarding to the
different litter components (Klamerus-Iwan et al., 2020; Coenders-Gerrits et al., 2020). It is interesting to note that not only the empirical $S_f$ falls within this range value, but also the observed $C_{fmax}$ (5.2 mm). The maximum daily evaporation observed was 2.0 mm day$^{-1}$ while the modeled values were 4.0 mm day$^{-1}$ and 5.5 mm day$^{-1}$ to Rutter and Gash models, respectively.

Others studies measured directly the forest floor evaporation (Schaap et al., 1997; Gerrits et al., 2007; Tsiko et al., 2012), or indirectly through different approaches (Bulcock and Jewitt, 2012a; Pitman, 1989; Putuhena and Cordery, 1996; Park et al.,
2010; Yang et al., 2018). To the best of our knowledge, there is no other study in Brazil with direct measurements of forest floor evaporation. Neto et al. (2012) applied the Rutter's model to a eucalyptus plantation in Brazil, and they simulated the forest floor interception using the mechanistic tank model. The authors found low forest floor interception value, 2.1 % and 2.4 % of $P_g$ to observed and modeled, respectively. Although also in a tropical region, the values are not comparable due to different litter compositions and measurement approaches. However, it is possible to compare the estimated values for the
same study area, based on laboratory test with the Cerrados's forest litter to obtain Cmax and Cmin (the last one conceptual similar to $S_f$), in which forest floor evaporation was slightly underestimated (i.e., 8.5% of $P_g$)(Rosalem et al., 2018).

### 3.6 Total interception

Total interception was obtained from the difference between rainfall and infiltration plus stemflow, while the modeled interception corresponded to the sum of canopy, trunk and forest floor evaporations. During calibration, the total interception
was overestimated by the Rutter model in 4.5 % (+29.9 mm) whereas the Gash model underestimated in -5.2 % (-34.0 mm). For the validation period, the observed interception corresponded to 41.4 % of total rainfall, and both models underestimated the total interception, -39.3 % (-210.5 mm) by the Gash model and -31.1 % (-166.7 mm) by the Rutter model. On average, the observed total interception was found to be 40.1 % of $P_g$, which is slightly higher than 26–36 % obtained by Gerrits et al. (2010) to a Temperate forest in Luxembourg.

As for the validation period the throughfall and infiltration were overestimated by studied models, it is expected that total interception had underestimated values. The results at monthly and biweekly time-scales indicate that the total modelled interception had a high normalized mean error (19–42 %) that may include the modeled errors regarding the different interception components. About to the monthly results, both models performed well. Rutter model had a slightly better


performance during the validation, with greatest NSE values and lowest RMSE and MBE values, while the Gash model had
the greatest R² values and better performance for the calibration period.

The total forest floor interception has a significant impact on the total interception process. In Cerrado *s.s.* forest, it corresponded to 35.8 % and 20.8 % of total interception, to calibration and to validation periods, respectively. This percentage difference may be explained by the short period to validation which did not include an entire dry season when the forest floor more likely dries out completely and the leaf-loss season happening.

Most forest interception studies focused on canopy and trunk interception since forest floor interception is more difficult for monitoring. Thus, there is no total interception measurements of tropical forest that include forest floor interception, that could be directly compared to those presented here. Besides, even for the same kind of forest, the total interception could have different ranges (Izidio et al., 2013; Junqueira Junior et al., 2019b; Medeiros and de Araújo, 2009; Sá et al., 2015b; Sari et al., 2015). It may occur due to different meteorological conditions (e.g., frequency of distribution, duration and intensity of the
storms) (Iida et al., 2017; Gerrits et al., 2010; Brasil et al., 2018), vegetation structure (Jiménez-Rodríguez et al., 2020; Sadeghi et al., 2016) and its seasonal changes (Fathizadeh et al., 2020; Hakimi et al., 2018; Sadeghi et al., 2018), to which sampling is subject.

The Gash and Rutter models are widely applied to forests across the world, with relative success to predict the interception (canopy and trunk) (Cui and Jia, 2014; Muzylo et al., 2009; Prasad Ghimire et al., 2017; Ringgaard et al., 2014; Sadeghi et al.,
2015). However, as pointed out by Linhoss and Siegert (2020), there are few studies that compare these models' performance and by whose accuracy at single-storm event scale are presented, the results indicate more error and poor relation. Our results suggest that the models could not model the higher interception daily-events, in agree with the pointed out by Linhoss and Siegert (2020). The Gash model presented a reasonable relation on the daily time-scale (calibration R² = 0.48 and validation R² = 0.57), unlike the Rutter model (R² values less than 0.02). However, it should be noted that observed total interception at
a daily time-scale is not taking into account the residence time of the different processes included, such as forest floor interception that took more than days to dry.

Regardless of the mentioned observed data issue, the Rutter model achieved, on average, higher daily interception values (1.36 mm day$^{-1}$) than the Gash model (1.23 mm day$^{-1}$). Nonetheless, the Rutter model had daily values below the maximum potential evaporation observed, unlike the Gash model that reached up to 12.0 mm in a day. This may have occurred to the Gash model
because the modeling evaporation considers the evaporation rate the same to all reservoirs. It was not a problem at a daily time-scale with the original Gash model due to the usually lower values of trunk evaporation, but for this adapted version, by including the forest floor reservoir, the inter-effect of the different interception components over the potential evaporation should be taken under consideration.

Although the evaporation from the different interception components is concomitant (regardless of the different rates), due to
the lower potential evaporation during the rain event and the great $S_c$ obtained, the evaporation processes in the adapted Rutter model were almost decoupled. As the canopy was expected to dry after a few hours, the reduced potential evaporation to the forest floor in the model should not cause much impact on the daily-based predicted values. However, the longer time to dry





the canopy caused a considerable delayed onset to forest floor evaporation. For 43 % (138 days) of rainy days, rain storms
felled during the night, which implies lower potential evaporation values, hence, poor daily prediction of the adapted model

was accentuated.

In spite of that, if we consider all the daily events (Fig. 4), the adapted Rutter model predicted well the cumulative-based total
interception with underestimation of 136.8 mm (-11.5 % of total interception) while Gash model underestimated by -244.6
mm (-20.5 %).

**Figure 4**. The relation between the monthly observed and modeled values of total interception by the adapted Rutter (grey) model and the

adapted Gash model (red) and the accumulate daily values for all monitored period.

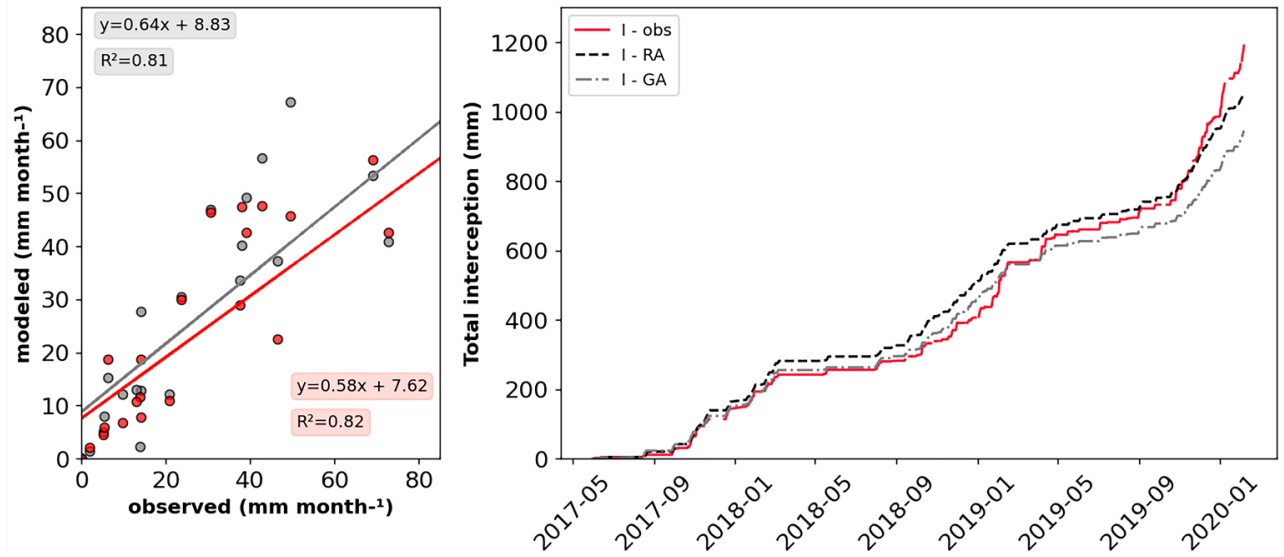

### 3.7 Temporal variability effect on the interception processes

The observed total interception is not a simple direct function of total rainfall. Biotic and abiotic factors interfere with this
process, like rainfall intensity, duration and distribution, and size and shape of water interception surfaces which directly affect

the storage capacities. The interception models applied to long data series can elucidate the role of the different factors on
responses on a seasonal scale.

To investigate the variability of the interception processes along the seasons, we applied the adapted models using seasonal
parameters (Table 6) obtained as the general parameters, based on the data of each season. The parameters, $Ds$, $b$, $Is$, $f$, $C_{cmax}$
and $C_{fmax}$ were maintained the same. Due to a few numbers of rainy days during the winter season, few events fulfilled the

conditions to be included in this analysis. Thus, the values of $S_c$ and $p$ could not be determined for winter. As the autumn and
winter seasons represent the dry period in the region, the autumn's values were maintained to the winter season.





Results indicate that the Gash model performed worse when seasonal coefficients were used, e.g., the total interception was more underestimated during both calibration and validation periods. The total errors of $T_f$, $F$, $E_c$ and were greater for the calibration and validation periods if seasonal coefficients are used (Table 7).

Unlike the Gash model, the Rutter model had similar tends regarding to accumulate values during the both periods. The total errors were slightly greater to $T_f$, $F$, $T_s$ and $E_f$ for both periods. However, there were better total estimates to $E_c$ and $I$, except to total interception during validation period, which increased the total error of 3 %. In general, the NSE values were similar, with greater differences to the stemflow, and the forest floor evaporation, which had NSE reduced up to 15 (dimensionless coefficient) for both calibration and validation periods.

**Table 6.** Seasonal ecohydrological parameters and the total number of rainy days used to its determination.

| Parameter | Summer | Autumn | Winter | Spring |
|---|---|---|---|---|
| $S_c$ (mm) | 2.9 | 2.0 | 2.0* | 2.5 |
| $p$ (unitless) | 0.35 | 0.52 | 0.52* | 0.45 |
| $S_f$ (mm) | 2.0 | 2.0 | 2.0 | 1.8 |
| $p_f$ (unitless) | 0.30 | 0.40 | 0.65 | 0.50 |
| $S_t$ (mm) | 0.020 | 0.030 | 0.023 | 0.010 |
| $p_t$ (unitless) | 0.003 | 0.002 | 0.003 | 0.002 |
| N° of rainfall events | 47 | 20 | 8 | 33 |

**Table 7.** Quantitative metrics and performance analysis of monthly estimations by the Rutter and Gash models with seasonal parameters during calibration (C) and validation (V) periods at monthly time-scale.

| Process | Period | Gash model adapted | | | | | | Rutter model adapted | | | | | |
|---|---|---|---|---|---|---|---|---|---|---|---|---|---|
| | | Total Error | RMSE | MBE | NMPE | NSE | R² | Total Error | RMSE | MBE | NMPE | NSE | R² |
| | | (mm) (%) | (mm month⁻¹) | | (%) | (unitless) | (unitless) | (mm) (%) | (mm month⁻¹) | | (%) | (unitless) | (unitless) |
| $T_f$ | C | +83.58 (11) | 20.42 | +10.30 | 19 | 0.78 | 0.98 | +29.44 (4) | 16.05 | +8.10 | 14 | 0.87 | 0.98 |
| | V | +149.70 (25) | 27.49 | +22.13 | 19 | 0.85 | 0.99 | +82.22 (14) | 15.31 | +10.38 | 9 | 0.96 | 0.98 |
| $F$ | C | +75.24 (7) | 12.33 | +1.71 | 14 | 0.91 | 0.93 | -28.00 (2) | 11.12 | -2.57 | 13 | 0.92 | 0.93 |
| | V | +229.65 (30) | 32.99 | +25.16 | 26 | 0.83 | 0.99 | +125.50 (16) | 19.72 | +13.60 | 15 | 0.94 | 0.99 |
| $T_s$ | C | -1.60 (46) | 0.21 | -0.18 | 46 | 0.47 | 0.75 | -1.92 (55) | 0.15 | -0.08 | 55 | 0.22 | 0.48 |
| | V | -1.57 (39) | 0.22 | -0.11 | 21 | 0.81 | 0.84 | -2.31 (58) | 0.28 | -0.24 | 29 | 0.50 | 0.78 |
| $E_f$ | C | -5.29 (2) | 6.34 | -0.22 | 37 | 0.21 | 0.63 | +12.99 (4) | 5.92 | +0.54 | 31 | 0.31 | 0.71 |
| | V | +0.14 (0) | 6.02 | -0.06 | 41 | 0.38 | 0.60 | +10.68 (9) | 4.63 | +1.10 | 26 | 0.63 | 0.87 |
| $E_c$ | C | -130.16 (35) | 14.26 | +4.56 | 25 | 0.61 | 0.65 | -54.87 (15) | 13.65 | +0.34 | 31 | 0.64 | 0.66 |
| | V | -158.92 (58) | 23.87 | -16.34 | 37 | 0.46 | 0.75 | -109.77 (40) | 15.82 | -8.54 | 25 | 0.76 | 0.79 |
| $I$ | C | -132.17 (20) | 11.72 | -4.50 | 19 | 0.81 | 0.84 | +1.53 (0) | 11.86 | +0.54 | 22 | 0.80 | 0.81 |
| | V | -256.35 (48) | 33.92 | -25.66 | 38 | 0.46 | 0.96 | -184.07 (34) | 24.12 | -15.30 | 25 | 0.71 | 0.90 |





The Rutter and Gash models' performances analyzed on the monthly time-scale for the entire dataset (Table 8), i.e., without

split it into calibration and validation periods, revels that seasonal coefficients did not cause considerable differences regarded to RMSE, MBE and NMPE values. Contrarily, the Gash model presented decreased performance to simulate the monthly total interception with seasonal coefficients.

**Table 8.** Quantitative metrics of monthly total interception estimated by the adapted versions of the Rutter and the Gash models from the entire dataset using seasonal and non-seasonal coefficients.

| Model | Coefficients | RMSE (mm month⁻¹) | MBE (mm month⁻¹) | NMPE (%) | NSE (unitless) |
|-------|-------------|-------------|-------------|------|------|
| Gash | Seasonal | 17.38 | -9.37 | 30 | 0.69 |
| | Non-seasonal | 14.64 | -8.82 | 27 | 0.78 |
| Rutter | Seasonal | 12.73 | -0.99 | 25 | 0.83 |
| | Non-seasonal | 13.16 | 0.63 | 26 | 0.82 |


Results of both models when the dataset is split into seasons (Table 9) confirm that the Gash model has in general, worse performance (lower NSE values) with seasonal coefficients. Nonetheless, the Rutter model had good performance in winter, autumn, and spring seasons, with decreased performance in summer, similar to the Gash model results with non-seasonal coefficients.

**Table 9.** NSE and R² (dimensionless) results of Rutter and Gash models by each season using the different model's coefficients.

| Model | Season coefficients | Summer | | Autumn | | Winter | | Spring | |
|-------|-------------|-----|-----|-----|-----|-----|-----|-----|-----|
| | | R² | NSE | R² | NSE | R² | NSE | R² | NSE |
| Gash | Seasonal | 0.68 | 0.09 | 0.99 | 0.52 | 0.74 | 0.66 | 0.73 | 0.37 |
| | Non-seasonal | 0.72 | 0.31 | 0.99 | 0.74 | 0.71 | 0.71 | 0.72 | 0.61 |
| Rutter | Seasonal | 0.59 | 0.33 | 0.97 | 0.69 | 0.71 | 0.69 | 0.74 | 0.73 |
| | Non-seasonal | 0.59 | 0.34 | 0.97 | 0.72 | 0.68 | 0.60 | 0.72 | 0.72 |

Winter $S_c$ value was lower than to other seasons as expected, since the Cerrado *s.s.* has markedly seasonal leaf fall peak in the dry season (include winter) (Alberton et al., 2019; Camargo et al., 2018). On the other hand, because the same reason, winter

$S_f$ greater than others $S_f$ was also expected. However, due to the few rainy days and the low rainfall intensity observed during winter (only 11 % of the rainy days and 7 % of the rainfall to the entire dataset occurred during winter), the $S_f$ estimated refers more to the observed water retained than the maximum static capacity of the forest floor to store water. Moreover, the decreased performance of both models in summer may indicate that this $S_c$ value was underestimated.

The forest floor evaporation and the total interception estimated by the Rutter model had similar values with seasonal and non-

seasonal coefficients (Fig. 5). The Gash model estimates had more differentation, with better estimates to the forest floor evaporation and worse to the total interception with seasonal coefficients. Bulcock and Jewitt (2012b) applied an adapted




version to the Gash model to model the canopy and forest floor evaporation of *Eucalyptus*, *Acacia*, and *Pinus* forests, being the forest floor evaporation estimated through drying curves equations. The relative errors of predicted forest floor interception by these authors were between 11 and 19 %, whereas our estimates resulted in maximum relative errors of 14 %, regardless

the model or coefficients and if calibration or validation periods.

**Figure 5**. Accumulated values of observed and modeled total interception and forest floor interception for all rain events monitored.

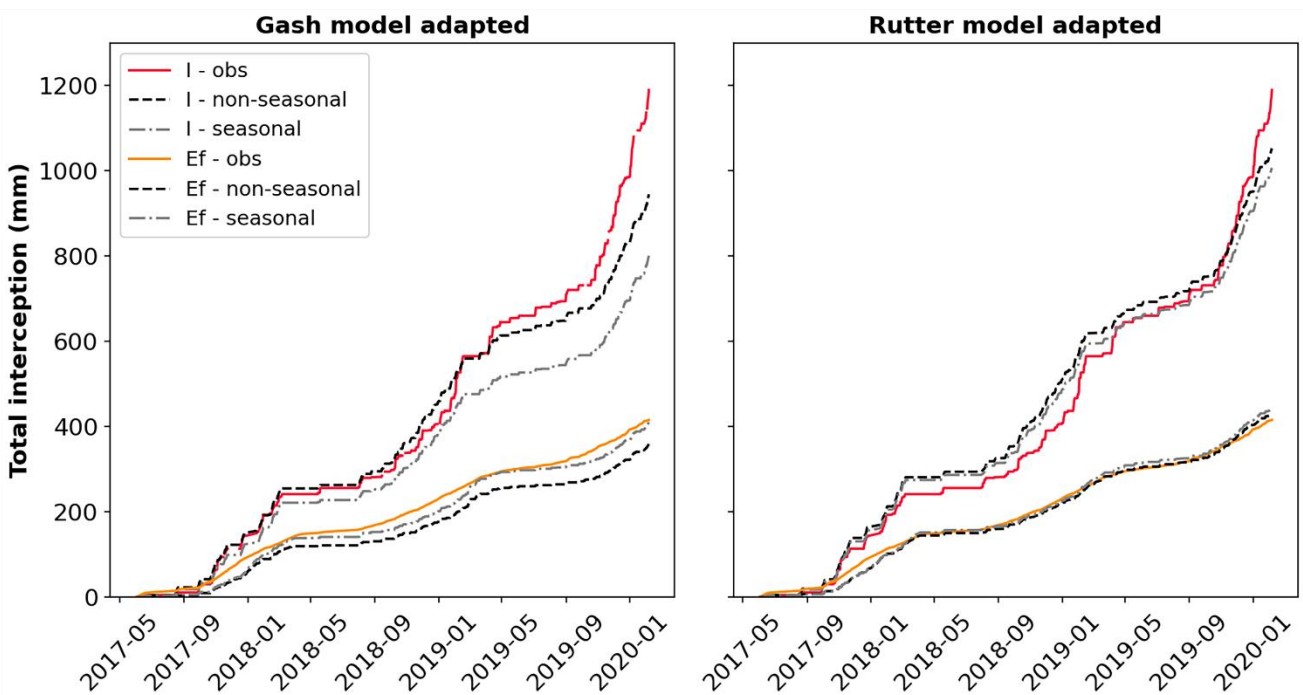

Taking into account that the modeled total interception includes modeling errors of the interception processes on the different

components resulted in good total estimates. The estimated values by the Gash model with non-seasonal coefficients, and by the Rutter model with seasonal or non-seasonal coefficients presented relative errors ≤ 20 %. Despite the Rutter model have presented lower relative error with non-seasonal coefficients to the total interception (11 % of underestimation) than with seasonal ones (15 % underestimation), it should be considered the improved canopy evaporation estimates obtained when seasonal coefficients were applied.

**Conclusions**

In Cerrado *s.s.* forest the total interception corresponds on average 40 % of total rainfall. Forest floor interception was, on average, 12 % of total rainfall, and 27 % of total interception. The Rutter and the Gash models were adapted to include this interception process, which both models poorly estimated on daily time-scale and reached at most a satisfactory performance





on a monthly basis (NSE ≤ 0.69). However, both models simulated well the accumulated forest floor evaporation for both,
calibration and validation periods, which were better estimated by the Rutter model (+0.9 % and +8.5 % of observed,
respectively) than by the Gash model (-14.2 % and -11.7 % of observed, respectively).

Regarding of total interception, both models had unsatisfactory performances at the daily time-scale with the worst
performance of the Rutter model which was mainly occasioned by the poor estimations of canopy and forest floor evaporations
at daily basis. This was confirmed as at biweekly time-scale both models had improved performances, with great and close
NSE values (from 0.56 up to 0.80). At monthly basis, models' performances were good, and if the dataset is not split on
calibration and validation periods, the Rutter model underestimates only 11 % (-136.8 mm of observed), in contrast to 20 %
of underestimation by the Gash model (-244.6 mm of observed). These results corroborate that these models present limitation
to simulate high rainfall events on an individual event-scale (Linhoss and Siegert, 2020). Despite the measurement errors, our
results point out that this modeling challenging can be greater for Cerrado forests since intense precipitation are more likely to
be observed; as we observed, with 10 days of canopy interception greater than 12 mm/day from 172 canopy interception
observations.

By using seasonal values for the models' parameters, the models had different responses. Whilst the Rutter model presented
at least improved performance in winter and spring seasons, the Gash model showed decreased performance for all seasons.
The improved performance of the Rutter model in winter and spring confirmed that the seasonal variations on forest structure,
mainly by leaf off from canopy during the dry season, affect the interception processes and that the models are able to respond
to such variations. Besides, it indicates that the storage and partitioning models' parameters as $S_c$ and $p$ may be less accurate
to summer and autumn, being non-seasonal values for the models' parameters may be more suitable for these two seasons. In
addition, as studies showed that vegetative phenological transitions in Cerrado (Alberton et al., 2019; Camargo et al., 2018)
do not strictly follow the seasons of the year, in future modelling applications will be interest take it under consideration.

Moreover, our results suggest that the evaporation during rainfall should be higher than the modelled estimations which caused
sometimes longer water residence's times on the canopy (more than a day). Thus, a better understanding of the energy fluxes
during storm events, as the controls on the downward heat flux and local horizontal advection (van Dijk et al., 2015),
considering also the vegetation structure, e.g., over- and understory energy fluxes (Jiménez-Rodríguez et al., 2020), could be
useful to improve the evaporation modelling.

Overall, the adapted models applied here are valuable to modelling the interception processes in Neotropical savannas like
Cerrado *s.s.* forests. The models had problems to simulate the evaporation processes at daily time-scale, but not to others
processes. Both models are suitable to estimate the total interception at monthly basis and could be used to inter annual
analysis, but for seasonal differences in Cerrado *s.s.* forests the Rutter model seems to be more appropriate.



**Appendix A**

**Table A1.** Equations to interception estimation for each rain day condition of the Gash model

| Rain day condition | Interception calculation |
| --- | --- |
| $m$ raindays (small storms insufficient to saturate the canopy) | $(1 - p - p_t) \sum_{j=1}^{m} P_{g,j}$ |
| $n$ raindays (storms large enough to saturate the canopy) | $n(1 - p - p_t)P'_g + \dfrac{\bar{E}}{\bar{R}} \sum_{j=1}^{n} (P_{g,j} - P'_g)$ |
| $q$ raindays (storms that saturate the trunks) | $qS_t$ |
| $(m + n - q)$ raindays (storms that do not saturate the trunks) | $p_t \sum_{j=1}^{m+n-q} P_{g,j}$ |

**Appendix B**

**Figure B1.** Cumulative distribution of observed rainfall and throughfall (mm) of the calibration and validation periods.

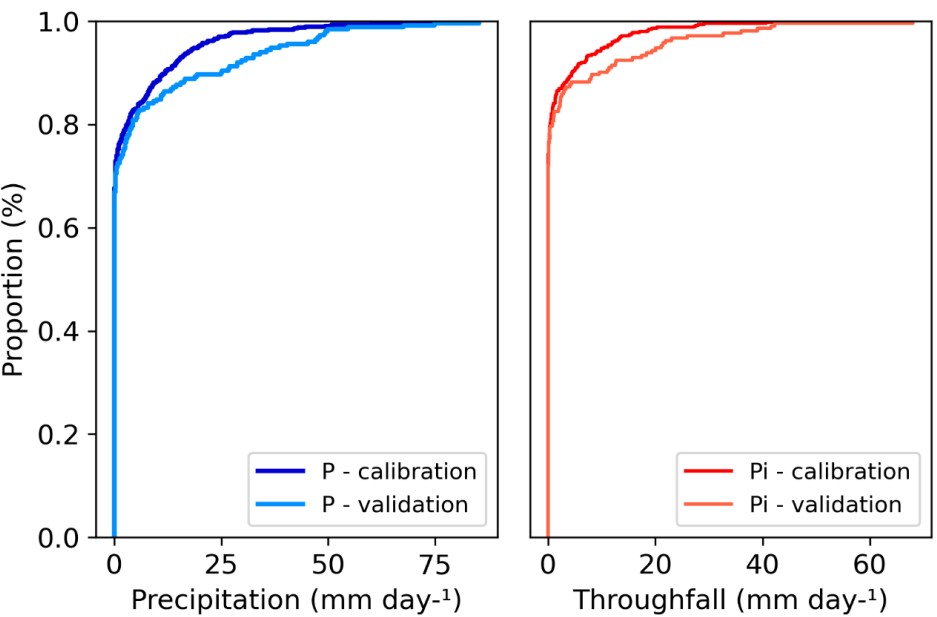





**Appendix C**

**Figure C1**. The relation between the daily values observed and modeled by the Rutter model adapted and the Gash model adapted to throughfall ($T_f$), infiltration ($F$), stemflow ($T_s$), canopy evaporation ($E_c$) and forest floor evaporation ($E_f$) and the accumulated values during the calibration period.






**Figure C2**. The relation between the daily values observed and modeled by the Rutter model adapted and the Gash model adapted to throughfall ($T_f$), infiltration ($F$), stemflow ($T_s$), canopy evaporation ($E_c$) and forest floor evaporation ($E_f$) and the accumulated values during the validation period.




*Data availability.* The dataset underlying this research are accessible from the following data repository link: http://www.hydroshare.org/resource/9134e6dc7cc94a999ee005966d0399f5

*Authors contributions.* LR, JAAA, EW designed the experiments, and LR and JAAA carried them out. LR, JAAA, and MC
analyzed the experiments' outcomes and discussed the results. LR, MC and SMMS prepared the paper with contributions from all co-authors.

*Competing interests.* The authors declare that they have no conflict of interest.

*Acknowledgements.* This study was supported by grants from the National Council for Scientific and Technological Development – CNPq (grant numbers 165010/2018-5 , 203252/2019-5); the São Paulo Research Foundation – FAPESP (grant number 2015/03806-1); and the Coordination of Improvement of Higher Education Personnel – CAPES (finance code 001, ad 88887.371140/2019-00). The authors acknowledge the graduate program in Hydraulics and Sanitary Engineering – PPGSHS (USP-EESC) – and the Water Management department at TU Delf for the scientific support. The authors would like also to
thank the Arruda Botelho Institute (IAB) for allowing the development of this study on its private land. and the editor and the anonymous referees for their useful comments, which substantially improved the paper.

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
