# Peer review of "Water partitioning in a Neotropical Savanna forest (Cerrado *s.s.*): interception responses at different time-scales using adapted versions of the Rutter and the Gash models"

_Hydrology and Earth System Sciences, 2022_

## Author Comment (AC2)

**Author's Response to Referee #2**

We would like to thank the anonymous Referee #2 for the detailed review and for the time spent reviewing our text. We replied to the referee's comments, which are incredibly useful to improve the quality of our manuscript. Note that the original referee's comments are identified as **R2C** and written in **bold**, and the authors' responses are labeled as **AR**. In addition, all comments are numbered (*e.g.*, **R2C-01**).

**[General comment]**
**This paper measured the interception processes including forest floor interception loss for a period of 32 months, and applied Rutter and Gash models. Although the methodology used in this study should be evaluated carefully, I recognized that the efforts to conduct measurements would be very intensive, and would agree the importance to understand interception phenomena in Cerrado region. However, the logic of this manuscript is not constructed well, so readers cannot understand what is the most important findings in this research. The main description seems to be the model performances for daily, biweekly and monthly basis. I understood the differences in performance among different time scales, but I do not think that the differences are important to clarify the interception process. The important thing must be what factors resulted in the differences and what processes affect the model performance. I would like to recommend that authors, at first, write a paper to show interception process in this site in detail based only on measurement data without applying any models. The interception loss in this site would be affected by rainfall characteristics (e.g., rainfall duration, rainfall amount, rainfall intensity, etc)? Or, other micrometeorological factors (e.g., wind speed, vapor pressure deficit, net radiation, etc) influence the interception process? Based on these knowledges, I believe that authors could develop suitable models to simulate the interception process. I hope my comments will help to do substantial revisions.**

**AR:** Thanks for your comments. As we said to referee 1 in R1C-06, our aim was to extent the existing Rutter and Gash models by adding the forest floor interception and verify the models' performance to simulate seasonal variations as it is pronounced for the Cerrado s.s. forest.
Our monitoring and model results show that the interception processes have different storage and evaporation times. The different residence times of water affect the evaporation modelling, which had impact to the forest floor evaporation modelling that takes several days (L349 – L353).
About the rainfall characteristics and meteorological factors that affect the interception processes, we mentioned some studies that have checked it (L385 – L392). Indeed, it would be interesting to present in separate paper more details about the effect of these factors on the interception processes in Cerrado based on our observed results.

**R2C-01: Could you add the LAI data? (line 92-93)**
AR: Yes, we will add information about the LAI of the study area based on remote sensing images.

**R1C-02: I cannot catch up how to calculate the spatially representative amount of throughfall. Is that the average of four Davis, five manual gutters and three automatic gutters? (line 100-119).**

AR: We used the average of all automatic sensors (four pluviographs, Davis Instruments, and three gutters connected to pluviographs P300, Irriplus Equipamentos - http://www.centev.ufv.br/incubadora/en-US/empresa/irriplus-tecnologia-e-manufatura-ltda). As the gutters could introduce more errors to low storm events, because of size of the gutters and its tipping bucket resolution (0.029 L), the gutters results were used when the rainfall intensity were greater than 3 mm/10min.

[Figure]

**R1C-03: Because the canopy in this site is discontinuous (line 93), I could expect that the spatial heterogeneity of throughfall would be very high. Please show the differences in throughfall amount measured by four Davis, five manual gutters and three automatic**

**gutters. The differences are related to the canopy openness? Also, a total of 12 measurements of throughfall are safely enough to obtain spatial representative value?**

AR: We intended to use all automatic throughfall measurements to have a representative average throughfall. However, the gutters (manual and automatic) had a bunch of clogging problems, and as we mentioned on R1C-02 because of its size of the gutters, only records during storm with intensity greater than 3 mm were considered to the average throughfall estimates.

The analysis of the pluviographs showed high values to the coefficient of quartile variation (CQV), between 79% and 90%, and high value of the coefficient variation for daily values (>170), which confirm the heterogeneity of the spatial distribution. The spatial heterogeneity of precipitation was also confirmed by the observed throughfall records (< 1.27 mm.day$^{-1}$) on days with no rain (24 days of observations). The effect of the heterogenous vegetation and canopy structure on throughfall distribution in the study was also observed through the 10 days which in the average throughfall was greater than the precipitation (average greater just 0.58 mm.day$^{-1}$ and maximum difference of 3.04 mm.day$^{-1}$). Despite the effects of the heterogenous spatial distribution of rainfall and heterogeneous vegetation the pluviographs and gutter presented good correlation with the daily precipitation, >0.94 and >0.87, respectively. The following figure show the daily values of the three pluviographs with more data in relation to daily precipitation.

[Figure]

**R1C-04: As Rosalem et al. (2018), published in Ecohydrology, pointed out, Davis gauge underestimates the inflow of water flux with increasing intensity (please see FIGURE 3 in**

Rosalem et al., 2018). I would like to confirm that authors applied the same correction to throughfall measurements in this paper. If not, application must be required.

AR: In Rosalem et al (2018), the tests were conducted with the LIDs, so the collect area and the resolution in millimeters are different that standard of the instrument. For the pluviographs used to monitor the throughfall directly (Davis gauges), the fabric specifications indicate errors ±4% of total rainfall for rain rates up to (50 mm.h$^{-1}$). For the study period, the maximum rainfall intensity was 57 mm.h$^{-1}$, so we could conclude that the measurement errors due to instrumentation is up to this value (±4 %).

**R1C-05: Three gutters are connected to three Davis gauge? If so, I am wondering that the one tip amount of 0.048 mm is too small to detect the correct amount. As Rosalem et al. (2018) showed, the underestimation by Davis gauge is relatively high. How many pulses generated by the gauge were recorded in 10-min intervals? The time between tips, equivalent to 600 second divided by accumulated pulse count, should be more than 1.0-1.5 second. Please note that, if authors used other rainfall gauge, similar issue exists and should be investigated.**

AR: No, each gutter is connected to a P300 gauge (Irriplus Equipamentos). Each tipping bucket has 29 mL, and considering the gutter area the resolution is 0.048 mm.

**R1C-06: How did authors calculate stemflow amount in the stand scale?**

AR: The stemflow was calculate dividing the total stemflow volume collected for each tree by its crown projected area. We will add this information on method's section.

(L119): *"All trees were selected according to the DBH. The depth (mm) of the stemflow monitored was obtained for each tree through dividing the stemflow volume collected (m³) by the crown projection area (m²)(Honda et al., 2015). The projection area was estimated considering a circular area through 3 diameter measurements with the stem at the center. More details about the equipment are given in the Table 1."*

**R1C-07: Three automatic collectors of stemflow is connected to Davis gauge? The same issue mentioned above, underestimation of inflow with increasing intensity, must be checked.**

AR: Automatic measurements of stemflow were carried out directing the stemflow collected from six trees, two tree per each rain gauge. As the intensity and volume of the stemflow are low for the study area (average values between 0.17 and 0.24 mm.day$^{-1}$, and maximum of 1.28 mm.dia$^{-1}$), with maximum stemflow of 1.05 mm.h$^{-1}$ (6.32 L.h$^{-1}$), the equipment error could be considered, as mentioned before on R2C-04, of ±4 %.

**R1C-08: Forest floor interception was measured by two LIDs (line 121-122). Please show the evidence indicating two LIDs could safely measure the spatial representative value of forest floor interception. This is very critical, because high spatial heterogeneity of throughfall could be expected from the disconnected canopy in this site.**

AR: It is undoubtable that more LIDs would better represent the forest floor interception of the study area. However, due to the cost of sensors and dataloggers, our data were based only these two LIDs.

Despite the few numbers of LIDs, the infiltration records monitored by these LIDs had good correlation among them (spearman coefficient equal 0.91, without zeros in the series), and the daily infiltration from both equipment have similar dispersion (Figure 1). The daily evaporation and storage of the forest litter measured by the LIDs presented good correlation, 0.73 and 0.77, respectively. The LID 2 usually stored and evaporated more than LID 1 (Figure 2), and that was due to the differences of the litter samples (weight and height), among other factors.

Figure 1. Relation between daily infiltration and precipitation.

[Figure]

Figure 2. Relation between the accumulated evaporation and storage of LID 1 and LID 2.

[Figure]

**R1C-09: Figure 2: The forest floor evaporation was calculated considering potential evaporation (Ep), but how did you calculate Ep above forest floor? Did you measure net radiation above the forest floor?**

**AR:** For the Rutter and Gash models the potential evaporation (Ep) is calculated through the Penman equation using the meteorological variables monitored in the monitoring station site 2, and when it was necessary the data from monitoring station 1 was used. The Ep calculated represents the threshold to the total evaporation for the models. Thus, the canopy and the trunk evaporation could not exceed this value, and for that the potential evaporation for the trunk evaporation process is reduced by the coefficient ϵ (Valente et al., 1997). As we intended to maintain the same assumptions of the original

models, the "residual" potential evaporation (Ep − Ec -Et) is applied on the forest floor evaporation modelling.

We suggested to change the sentence in L348 in response to referee 1 in R1C-34, and as said on that comment, we add the forest floor interception to the Rutter and Gash models changing as minimum was possible the original models' assumptions. With that we could see the possible issues for including the forest floor in the model, which could help to improve the measurements needed (e.g., energy budget at soil level in the forest) and after the modelling.

**R1C-10: Please add the description to explain how to calculate the aerodynamic conductance above canopy and forest floor (line 214-215).**

**AR:** We did not calculate the aerodynamic resistance. As said in L145, the Ep was calculated through Penman equation. When the surface resistance in the Penman-Monteith equation is equal to zero, the result is equal when Penman equation is used.

**R1C-11: How did you obtain the interception ratio of 33%? (line 253) Throughfall was described as 70-72% (line 228), so 33% interception is too large. There is a description of 40% interception in conclusion section (line 481). Maybe the target rainfall events are different among parts, but it is difficult to understand.**

**AR:** The average value 33% (L253) was calculated using the observed daily data of the entire series. In L259-260 we showed the values for the calibration (33%) and validation (27%) periods. We verified the average value as said in L260-264, which indicated that the process would be equal to 28%. So we understand that it occurred due to the method used (fixed throughfall gauges), also because the period considered in the calculus since the canopy cover vary with seasons in Cerrado *sensu stricto* (Alberton et al., 2019).

**R1C-12: I cannot understand why Ec is calculated as the difference Pg and the sum of F, Ts and Ef. I recommend that basic equation showing rainwater balance should be added in the M&M section.**

**AR:** We use the basic equation as mentioned in L253. We calculated Ec using the other equation to verify the values since for the other equation the direct measurements of the forest litter interception and infiltration with the LIDs could be used.

**R1C-13: Discussion about the stemflow is not directly related to this paper (line 300-313). If authors show the data of canopy structure, bark, and so on, it is useful. Unfortunately, the current MS did not include any data, so I recommend to remove this part.**

**AR:** We included more information about stemflow in Cerrado to show the state of research of this subject. As there is no space to include our monitoring results and modelling at same paper, we can remove L300-313.

**R1C-14: I felt that the much sentences in the current MS are related to model performance (line 237-245, 268-273, 293-299, 320-328, 335-342, 368-380, 435-454, Table 6, 7, 8, 9). Similar descriptions are found among parts, so I recommend reconstruction of the logic. In my**

opinion, it would be better that "discussion" should be separated from the "result" section. Then, descriptions of model performance should move to discussion section, and more concise discussion is recommended. Rather than differences in model performance, the reason for the difference and factors affecting it are more important to understand the interception process in this site.

**AR:** We understand that separate the results from the discussion would make more difficult for the reader to follow the discussion, as there are different processes and different models.

About the focus on the models' performance, we have to focus on that since our main purpose was to extent the models by adding the forest floor interception and verify the models' performance to simulate seasonal variations as it is pronounced for the Cerrado *sensu stricto* forest.

**R1C-15: Looking at appendix C, there are high correlations between observation and model output for throughfall and stemflow. However, the correlation for interception loss is very low. Could you explain this point?**

**AR:** Based in our finds the main problem with the models is on the evaporation. The potential evaporation during the rainfall is very low mainly due to low income radiation values during the storms. Besides it, the potential evaporation is partitioned instead to be modelling separately between above and below the canopy and because that the evaporation processes in the study were almost decoupled (L409-415).

**[Technical corrections]**

**R1C-16: Line 98: "average PAR (photosynthetic active radiation) of 1041.8 ± 427.4 μmol.m-2.s-1", I cannot understand the duration of average.**

**AR:** It is a common unit for PAR measurements (Custódio et al., 2021). This information was obtained by (Reys et al., 2013), but, as commented in response to referee 1 (R1C-08), it is an additional information that can be removed.

**R1C-17: Line 199-200: "From June of 2019 to May of 2019, we used to calibrate our Rutter and Gash models, while the second part, from June of 2019 up to of 07th February of 2020, .."Please check the consistency of months. The current description is strange.**

**AR:** This information was wrong. This error was also mentioned by the referee 1 in R1C-23. It was supposed to be "From June of 2017 [...]", such as the monitoring period mentioned in L100.

**R1C-18: Figure 3, y-axis title of the upper panel: "Potencial" should be "Potential".**

**AR:** Thanks for mention. We will correct it.

**R1C-19: Figure 3, lower panel: Is this net radiation? Solar radiation was measured at 2 m height (Table 1). More than 600 W m-2 value is too high for solar radiation above the forest floor. Please check carefully.**

**AR:** Solar income radiation is monitored in IAB 1 monitoring station. It is a standard meteorological station, which means that the most of sensors are set 2 m above the grass field in an open area. Higher solar radiation income occurs during summer season in the study region (Cabrera et al., 2016).